# Prediction of Diabetes and Prediabetes among the Saudi Population Using a Non-Invasive Tool (AUSDRISK)

**DOI:** 10.3390/medicina60050775

**Published:** 2024-05-07

**Authors:** Ayoub Ali Alshaikh, Faisal Saeed Al-Qahtani, Hassan Misfer N Taresh, Rand Abdullah A Hayaza, Sultan Saeed M Alqhtani, Sarah Ibrahim Summan, Sultan Abdullah Al Mansour, Omar Hezam A Alsultan, Hassan Yahya M Asiri, Yazeed Mohammed S Alqahtani, Waleed Khaled A Alzailaie, Ahmed Abdullah A Alamoud, Ramy Mohamed Ghazy

**Affiliations:** 1Family and Community Medicine Department, College of Medicine, King Khalid University, Abha 61423, Saudi Arabia; alashaikh@kku.edu.sa (A.A.A.); drfaisalqahtani@gmail.com (F.S.A.-Q.); hmnt5005@gmail.com (H.M.N.T.); randalhayaza@gmail.com (R.A.A.H.); sultanalkasi@icloud.com (S.S.M.A.); saraibraheem@hotmail.com (S.I.S.); amo0ory505@gmail.com (O.H.A.A.); dr.hassan3.201@gmail.com (H.Y.M.A.); yyqq2030@gmail.com (Y.M.S.A.);; 2Asser Center Hospital, Ministry of Health, Abha 62521, Saudi Arabia; bshri.505@gmail.com; 3Tropical Health Department, High Institute of Public Health, Alexandria University, Alexandria 61421, Egypt

**Keywords:** diabetes mellitus, prediabetes, Saudi Arabia, prediction, AUSDRISK

## Abstract

*Background and Objectives*: Screening for type 2 diabetes mellitus (DM2) aims to identify asymptomatic individuals who may be at a higher risk, allowing proactive interventions. The objective of this study was to predict the incidence of DM2 and prediabetes in the Saudi population over the next five years. *Materials and Methods*: The study was conducted in the Aseer region through August 2023 using a cross-sectional survey for data collection. A multistage stratified random sampling technique was adopted, and data were collected through face-to-face interviews using the validated Arabic version of the Australian Type 2 Diabetes Risk Assessment Tool (AUSDRISK). *Results*: In total, 652 individuals were included in the study. Their mean age was 32.0 ± 12.0 years; 53.8% were male, 89.6% were from urban areas, and 55.8% were single. There were statistically significant differences between males and females in AUSDRISK items, including age, history of high blood glucose, use of medications for high blood pressure, smoking, physical activity, and measurements of waist circumference (*p* < 0.05). Based on AUSDRISK scores, 46.2% of the included participants were predicted to develop impaired glucose tolerance within the coming five years (65.8% among females vs. 23.6%), and 21.9% were predicted to develop DM2 (35.6% among males vs. 6.0% among females); this difference was statistically significant (*p* = 0.0001). *Conclusions*: Urgent public health action is required to prevent the increasing epidemic of DM2 in Saudi Arabia.

## 1. Introduction

The prevalence of diabetes mellitus (DM) is experiencing a significant global increase, especially in the Middle East and North Africa (MENA), owing to factors such as urbanization, aging of the population, and lifestyle changes [1,2]. The International Diabetes Federation (IDF) estimates that 537 million adults aged 20–79 years will have diabetes by 2021, representing approximately one in every ten people. This figure is expected to rise to 643 million by 2030 and then to 783 million by 2045. More than three out of every four people with diabetes live in low- and middle-income countries, highlighting the global burden of this condition in less-prosperous regions [3]. In 2021 alone, DM was responsible for 6.7 million deaths, equivalent to one person losing their life every 5 s due to the disease. The economic burden of diabetes is significant, with health expenditure reaching at least 966 billion USD by 2021. This marks a notable 316% increase over the past 15 years, underscoring the financial strain it imposes on the healthcare system. Furthermore, 541 million adults have impaired glucose tolerance (IGT), a condition that puts them at a high risk of developing type 2 diabetes (DM2) [3].

DM2 accounts for 90–95% of all diagnosed cases of diabetes, while type 1 diabetes accounts for 5–10% [4]. The development and progression of DM2 are influenced by several modifiable and non-modifiable risk factors. Obesity, physical inactivity, poor diet, high blood pressure, and smoking are all possible risk factors. Non-modifiable risk factors, on the other hand, include age, family history, ethnicity, and genetic predisposition [5]. IDF emphasizes lifestyle modifications such as increased physical activity and the adoption of a healthy diet as effective measures to prevent or delay the onset of DM2 [6].

Approximately 73 million people in the MENA region live with DM. In 2021, diabetes affected 17.7% of the total adult population in Saudi Arabia alone, amounting to approximately 4,274,100 cases [7]. A comprehensive epidemiological health study was conducted in Saudi Arabia, focusing on adults aged 30–70 years. Among the 16,917 survey participants, 4004 were diagnosed with DM, representing approximately 23.7% of the population [8]. However, higher prevalence rates of DM have been observed among Saudis, ranging from 26.0% to 61.8% [9,10].

These figures emphasize the importance of effective preventive strategies and enhanced care management to address the country’s increasing DM burden. One of these techniques is DM screening, which attempts to identify asymptomatic patients who may be at risk, thus allowing for preventive therapy [11]. For a long time, DM screening depended on invasive, inconvenient, and costly approaches, such as blood samples for fasting plasma glucose (FPG), the 2-h oral glucose tolerance test (OGTT), or measurements of glycated hemoglobin (HbA1c) [12]. However, researchers in many countries have devised non-invasive risk assessment scoring methods that are more practical, less time-consuming, and cost-effective for detecting DM2 [13]. These risk scores incorporate the most effective behavioral and biological risk factors into a scaled instrument for dual screening and prediction functions. Several risk scores, including the American Diabetes Association Diabetes Risk Test (ADADRT), the Finnish Diabetes Risk Score (FIN-DRISK), the Canadian Diabetes Risk Questionnaire (CANRISK), the German Diabetes Risk Score (GDRS), and the Australian Type 2 Diabetes Risk Assessment Tool (AUSDRISK), have been developed and used around the world [14,15].

A team of Egyptian researchers translated AUSDRISK into Arabic and assessed its reliability and validity among Egyptians. The recalibrated Arabic AUSDRISK was shown to be a simple, reliable, and valid tool for predicting DM risk, suggesting its use in mass public screening to reduce the disease burden and health costs [16]. Therefore, its use in opportunistic mass public screening is highly recommended, owing to its cost-effectiveness. By employing this screening tool, it is possible to preemptively identify individuals at risk of developing DM2, which can significantly reduce the disease burden and associated health expenditure. With early detection and targeted preventive measures, the burden of DM2 can be mitigated, leading to better health outcomes for the population [17].

The objective of this study was to predict the incidence of DM2 and prediabetes in the Saudi population over the next five years using the valid Arabic version of the AUSDRISK tool. Using AUSDRISK, researchers aimed to gain insight into the future incidence of DM2, thus facilitating the development of effective preventive measures and interventions to address the health challenges posed by DM in Saudi Arabia.

## 2. Materials and Methods

### 2.1. Study Setting and Study Duration

This study was conducted in the Aseer region of Saudi Arabia, one of the country’s administrative regions, situated in the southwestern part of Saudi Arabia, with Abha as its capital. This study was conducted in August, 2023.

### 2.2. Study Design and Sampling Technique

This study used a cross-sectional survey for data collection utilizing a multistage stratified random sampling technique. In the Aseer region, which consists of 17 governorates, three were randomly selected. One health center was randomly included in each selected governorate. The selection of apparently healthy adults visiting these centers was conducted using a systematic random sampling method.

### 2.3. Sample Size and Study Population

The minimum required sample size, as determined using G*Power software version 3.1, Franz Fuel, Germany, to investigate the prevalence of undiagnosed DM2 among the Saudi population was 318, based on the following assumptions: an expected prevalence of diabetes of 17.7%, sensitivity of AUSDRISK set to 81.3%, specificity set to 57.7%, an alpha error of 0.05, a non-response rate of 5%, and a precision of 10%. The sample size was doubled to compensate for the stratification based on sex. The study included apparently healthy Saudi adults without DM, aged 18 years or older, who resided in the Aseer region.

### 2.4. Data Collection

Data were collected through face-to-face interviews with respondents. The first section of the questionnaire focused on collecting information on various socioeconomic factors. This included capturing the participants’ age, sex, residence (rural or urban), marital status (divorced, married, single, or widowed), and education (secondary, university, or postgraduate levels). Participants’ monthly income levels were categorized into specific ranges from below 5000 Saudi Riyal (SAR) to above 20,000 SAR, and their occupation was recorded as do not work, governmental sector, private, or retired. Finally, participants were categorized by nationality as Saudi or non-Saudi. Respondents were asked about associated comorbidities, such as hypertension, thyroid issues, renal disease, liver disease, ischemic heart disease, chronic obstructive airway disease, cerebrovascular disease, and autoimmune disease. The second section included the validated Arabic version of the AUSDRISK tool [18], which assesses nine risk factors: age, sex, family history of diabetes, history of high blood glucose, hypertension, smoking, fruit and vegetable intake, physical activity, and waist circumference (Table 1). The sensitivity and specificity of the Arabic version of AUSDRISK at a cut-off point of 13 were 86.11% and 73.35%, respectively, with a 0.887 (95% confidence interval (CI): 0.824–0.95) area under the curve (AUC), *p* < 0.001. Sensitivity and specificity for the detection of abnormal hyperglycemia levels were at a cut-off point of 9 with an AUC of 0.767 (CI: 0.727–0.807), *p* < 0.001 [18].

Anthropometric measurements: To measure waist circumference, assessors gently surrounded the waist of the respondent with measuring tape, positioning it at the level of the top of the hip bone. At a relatively large waist circumference (102 cm for men and 88 cm for women), specific cutoff points have been established to identify risk. Ensure that the tape runs parallel to the ground and forms a complete circle around the trunk. Body mass index (BMI) was calculated using the following formula: BMI = weight (kg)/height (m)^2^. To calculate BMI, the body weight in kilograms was divided by the height squared in meters. The BMI categories were as follows: BMI below 18.5 is classified as underweight, while BMI ranging from 18.5 to less than 25 was considered within the healthy weight range. A BMI between 25.0 and less than 30 indicates that the individual is overweight, and a BMI of 30.0 or higher denotes obesity [19].

### 2.5. Study Outcome

The primary objective of this study was to determine the expected incidence of diabetes and prediabetes among apparently healthy adults over the next five years. In addition, this study aimed to identify the determinants associated with these conditions.

### 2.6. Ethical Consideration

Ethical approval was obtained from the Ethics Committee of the King Khalid University, Abha, Saudi Arabia (ECM # 2023-401). To ensure ethical conduct, all participants provided written informed consent before participation. The study strictly adhered to the ethical guidelines of the Declaration of Helsinki, ensuring the protection of the rights, safety, and confidentiality of participants, while upholding the highest ethical standards in research.

### 2.7. Statistical Analysis

Data were analyzed using R software version 4.2. Numerical variables were summarized using the mean and standard deviation. In cases where the data exhibited skewness, the median and interquartile range (IQR) were used. Categorical variables were presented as numbers and percentages. The association between the outcome and different categorical variables was assessed using the chi-square test. The Mann–Whitney U test was used to compare non-normally distributed numerical variables. The significance level (*p* value) for all statistical tests was set at 0.05, indicating statistical significance at *p* < 0.05.

## 3. Results

Table 2 describes the demographic and socioeconomic variables of the 652 individuals studied. The mean age of the participants was 32 ± 12 years, ranging 18.0–92.0 years; 53.8% of the studied participants were male, 89.6% were from urban areas, and approximately three-fifths of the participants were single (55.8%), followed by married (39.6%). Regarding education, 23.0% attained secondary, 67.2% university, and 7.5% postgraduate degrees. The largest group in terms of earnings was below 5000 SAR (41.0%), followed by 5000–10,000 SAR (30.2%). Nearly two-fifths of the participants (38.7%) were unemployed, followed by those working in the government sector (36.5%), with 19.5% working in the private sector. The majority of the participants were Saudi (92.9%), with a smaller percentage being non-Saudi (7.1%).

The mean BMI was 26.0 ± 8.80 (range 16.0–22.0). Hypertension was reported in 5.7% of the participants, while thyroid disease was reported in 2.9%. The prevalence rates of renal and liver diseases, ischemic heart disease, chronic obstructive airway disease, cerebrovascular disease, and autoimmune diseases range from 0.2% to 1.8%. A significant proportion of participants (49.2 %) reported a family history of DM (Table 3).

Table 4 shows that there were statistical differences between men and women in AUSDRISK items, including age (χ^2^ = 21.96, *p* = 0.001), previous history of high blood glucose (sugar) (χ^2^ = 7.527, *p* = 0.006), taking medication for high blood pressure, which was also significant across gender (χ^2^ = 16.178, *p* = 0.0001), smoking (χ^2^ = 139.371, *p* = 0.0001), physical activity [χ^2^ = 12.06, *p* = 0.001], and waist circumference (χ^2^ = 27.002, *p* = 0.0001).

Figure 1 shows that 21.9% (143) will develop diabetes, 35.6% among males vs. 6.0% among females (χ^2^ = 81.38, *p* = 0.0001). Similarly, 46.30% (302) of the participants will develop impaired glucose tolerance in the next five years as they scored 9 or more on AUSDRISK (65.8% among women vs. 23.6% among men); this difference was statistically significant (χ^2^ = 114.50, *p* = 0.0001) (Figure 2).

Several sociodemographic factors were associated with an AUSDRISK score of 13 or higher, including marital status, education, income, residence, and occupation. Married participants had a higher AUSDRISK score than single participants (66.4% vs. 32.0%; *p* < 0.001). Regarding educational level, 72.7% of those with AUSDRISK < 13 had a university degree, compared to 53.8% of those with AUSDRISK ≥ 13 (*p* < 0.001). According to the results, 45.4% of individuals with an AUSDRISK score of less than 13 had a monthly income of less than 5000 SAR, compared to only 25.2% of those with an AUSDRISK score greater than or equal to 13. This difference was statistically significant (*p* < 0.001). Significant differences were also observed between residences and occupations. There was also a significant association between having chronic diseases, like ischemic heart disease and hyperlipidemia, and a high AUSDRISK (Table 5).

## 4. Discussion

DM2 is a metabolic disorder characterized by insulin resistance and pancreatic β-cell dysfunction. The role of inflammation in insulin resistance can be traced to the integration of metabolism and innate immunity via nutrient-sensing pathways that are mutually related to pathogen-sensing pathways [20,21].

### The Main Findings of the Study

In this study, we aimed to predict the incidence of prediabetes and DM2 among the apparently healthy Saudi population in the Aseer region using the validated noninvasive Arabic version of AUSDRISK. Interestingly, more than two-fifths of the participants were predicted to develop either prediabetes or DM2 in the coming five years. The AUSDRISK score was higher among males than among females, with a higher prevalence of undiagnosed hyperglycemia. Several factors were associated with undiagnosed hyperglycemia, including marital status, educational level, income, residence, occupation, ischemic heart disease, and hyperlipidemia.

Tools to predict the incidence of diabetes: Various risk models, commonly referred to as risk scores, have been developed to identify cases of DM2. Although some are effective in detecting undiagnosed (prevalent) DM2 cases [22], others predict the onset of new (incident) DM2 cases, including the Finnish Diabetes Risk Score (FINDRISC) [23], Peruvian Risk Score [24], and AUSDRISK. In this study, we used the AUSDRISK tool to predict the incidence of DM2 in the Saudi population. AUSDRISK proved to be a straightforward, cost-effective, and efficient approach to assess diabetes risk compared to blood glucose testing [25]. Ten different medical, lifestyle, and demographic factors were incorporated into the AUSDRISK tool, which was created for use within an Australian community to predict the 5-year risk of DM2. The Australian Government Department of Health and Aging established the AUSDRISK tool in July 2008, and qualified patients can receive Medicare reimbursements [15]. In this study, we used the validated Arabic version, which consists of nine questions. The ethnicity question was omitted during the validation process. However, the tool still shows high performance in the prediction of DM2 incidence.

Predicted incidence of diabetes and prediabetes: The frequency of diabetes has increased rapidly in Saudi Arabia, from 7% in 1989 to 32% in 2009 [26]. Moreover, the prevalence of undiagnosed DM2 is high; nearly one-third (36.1%) of the apparently healthy population in Alqunfudah, Saudi Arabia, was diagnosed with DM2, and 28.3% had impaired fasting glucose [27]. Interestingly, in this study, we predicted that 46.2% will develop impaired glucose tolerance within the next five years since they scored above 9 on UDSRISK, and 21.9% will develop DM2 because they scored 13 or above on AUDRISK. Our findings support a previous study that predicted that the prevalence of DM2 in Saudi Arabia would increase from 32.8% in 2015 to 45.36% in 2030 [28]. These findings indicate an alarmingly high incidence of DM, as well as a change in the prevalence of DM over time. According to the IDF, the prevalence of diabetes in Saudi Arabia is among the top ten countries among people aged 20–79 [29]. The high prevalence of DM highlights the urgent need to adopt novel public health strategies to effectively manage this disease in Saudi Arabia. These strategies may include community-based screening programs [27,30] and focused interventions aimed at increasing public awareness of diabetes mellitus and its consequences [31].

Factors associated with the development of diabetes: We found that males were more likely to have DM2 than females, and the same was true for impaired glucose levels. Several studies conducted in Saudi Arabia have found a higher frequency of DM in males [8,32]. The high prevalence of obesity among males may be attributed to unhealthy dietary habits [33] that result in an increased risk of obesity [34].

We found that the adoption of healthier lifestyles in the form of physical activity and non-smoking was more common among women than among men. According to the 2014 Surgeon General’s Report, active smokers face a 30–40% higher risk of DM2 than non-smokers [35]. In addition, smoking can exacerbate the challenges associated with managing diabetes and regulating insulin levels. Nicotine diminishes the effectiveness of insulin, necessitating that smokers rely on increased insulin doses to stabilize their blood sugar levels [36]. This underscores the importance of emphasizing smoking cessation as a crucial public health strategy for controlling diabetes worldwide.

Elevated BMI and decreased physical activity were both independently associated with an increased risk of developing DM. Obese people had a nine-fold higher risk than normal-weight people. Physically active individuals had almost two-thirds of the risk of diabetes compared to those who were inactive. Obese and sedentary men had a 17-fold increased risk of diabetes when BMI and physical activity were combined [37].

Several sociodemographic factors were associated with higher AUSDRISK scores, including education, income, residence, occupation, and hyperlipidemia. Similarly, a study conducted in the Eastern Province between 2004 and 2005 found some sociodemographic factors associated with an increased prevalence of DM2. Being widowed (39.1%), unemployed (31.9%), and uneducated (32.3%) was associated with the development of DM2 [38]. Another study conducted by Al-Zahrani and Aldiab [39] found that being older, married, obese, a smoker, or less educated significantly increased the risk for both prediabetes and diabetes.

Implications of this study: This study highlights the sex disparities regarding risk factors associated with DM2, such as smoking, physical activity, and waist circumference, emphasizing the need for tailored prevention strategies. This study also predicts future DM2 and prediabetes based on the AUSDRISK scores and highlights the feasibility of this risk assessment tool. This research further explains the complex interplay of factors that influence diabetes risk, including associations between AUSDRISK scores and sociodemographic factors such as marital status, education, income, residence, occupation, and chronic diseases. Overall, this study underscores the necessity for holistic approaches to diabetes prevention and management, considering diverse sociodemographic and health-related factors.

Strengths and limitations: To our knowledge, this is the first study conducted in Saudi Arabia using a non-invasive tool to predict the incidence of diabetes. Second, we used a multistage stratified random sampling technique to obtain a representative sample from the Aseer region, considering the demographic variations in different governorates. Finally, a validated questionnaire was used to strengthen the internal validity of the study. However, the study used a cross-sectional design, which limits its ability to establish causal relationships between risk factors and the development of DM2 or prediabetes. Despite the stratified sampling technique, there may still be some degree of sampling bias, as participation in the study was based on individuals visiting health centers. This may not capture the experiences of those who do not seek healthcare services. The study relied on self-reported data, which can be subject to recall or social desirability bias, where participants may provide answers that they perceive as more socially acceptable. This study was limited to the Aseer Region of Saudi Arabia. These findings may not be fully representative of the entire Saudi population, which exhibits regional variations in lifestyle and health. Finally, in this study, we did not focus on certain risk factors for the development of DM2, such as stress and dietary habits. Future studies should assess the value of incorporating these factors into the AUSDRISK tool to increase its diagnostic performance.

## 5. Conclusions

These findings indicate that a substantial proportion of the population studied is at risk of developing impaired glucose tolerance and DM2 within the next five years. These risks are significantly higher among men than among women. These findings underscore the urgent need for targeted public health interventions to address the escalating diabetes epidemic in Saudi Arabia. Healthcare authorities and policymakers must prioritize comprehensive diabetes prevention and management strategies, focusing on risk assessment, lifestyle modifications, health education, and access to healthcare services, particularly for high-risk groups. Early detection and proactive interventions are critical to mitigate the potential health and economic burdens associated with diabetes in the Saudi population.

## Figures and Tables

**Figure 1 medicina-60-00775-f001:**
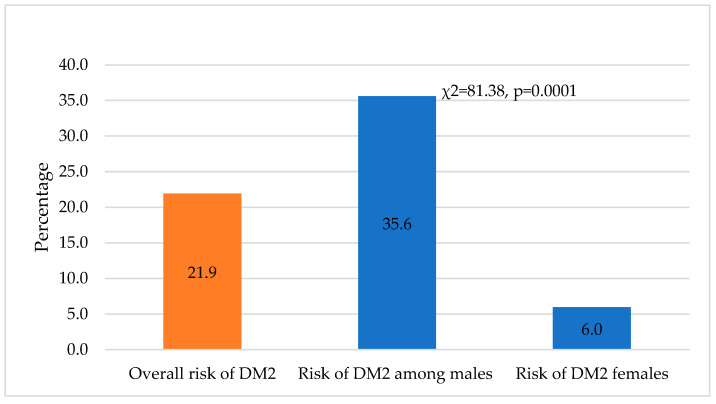
Risk of developing diabetes mellitus in the coming five years among all study participants and between males and females.

**Figure 2 medicina-60-00775-f002:**
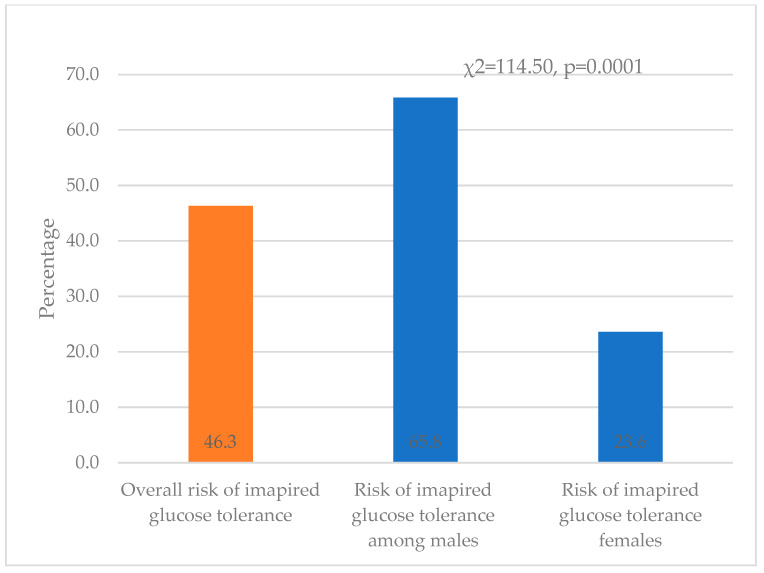
The risk of developing impaired glucose tolerance in the coming five years among all study participants and between males and females.

**Table 1 medicina-60-00775-t001:** Items of the validated Arabic version of AUSDRISK.

	Question		Points
1	Age		
	Less than 35 years		0
	35–44 years		2
	45–54 years		4
	55–64 years		6
	65 years or older		8
2	Gender		
	Female		0
	Male		3
3	Has either of your parents or siblings been diagnosed with diabetes (type 1 or type 2)?		
	No		0
	Yes		3
4	Have you ever had high blood sugar levels (e.g., during a health checkup, illness, or pregnancy)?		
	No		0
	Yes		6
5	Currently, are you taking medication for high blood pressure?		
	No		0
	Yes		2
6	Currently, do you smoke cigarettes or any other tobacco product daily?		
	No		0
	Yes		2
7	Typically, how often do you consume vegetables or fruits?		
	Every day		0
	Not every day		1
8	On average, do you believe you engage in physical activity for at least 2.5 h per week (e.g., 30 min daily for 5 days or more per week)?		
	Yes		0
	No		2
9	Waist measurement at the bottom of the ribs (usually at the level of the navel and while standing):		
	Waist measurement (cm)		
	Men	Women	
	Less than 102 cm	Less than 88 cm	0
	102–110 cm	88–100 cm	4
	More than 110 cm	More than 100 cm	7

**Table 2 medicina-60-00775-t002:** Sociodemographic characteristics of the study population.

Studied Variables	Overall (N = 652)	
Age	Mean (SD)	32.0 ± 12.0
	Median [Min, Max]	28 [14, 92]
Sex	Female	301 (46.2%)
	Male	351 (53.8%)
Residence	Rural	68 (10.4%)
	Urban	584 (89.6%)
Marital status	Divorced	24 (3.7%)
	Married	258 (39.6%)
	Single	364 (55.8%)
	Widow	6 (0.9%)
Education	Secondary	150 (23.0%)
	University	453 (67.2%)
	Postgraduate	49 (7.5%)
	Below 5000 SAR	267 (41.0%)
Income (month)	5000–10,000 SAR	197 (30.2%)
	10,000–15,000 SAR	104 (16.0%)
	15,000–20,000 SAR	55 (8.4%)
	More than 20,000 SAR	29 (4.4%)
Occupation	Do not work	252 (38.7%)
	Governmental sector	238 (36.5%)
	Private	127 (19.5%)
	Retired	35 (5.4%)
Nationality	Non-Saudi	46 (7.1%)
	Saudi	606 (92.9%)

**Table 3 medicina-60-00775-t003:** Medical history of study population.

Studied Variables	Mean (SD)	
Body mass index	Median [Min, Max]	26.0 [16.0, 22.0]
Hypertension		37 (5.7%)
Thyroid		19 (2.9%)
Renal		4 (0.6%)
Hepatic disease		1 (0.2%)
Ischemic heart disease		12 (1.8%)
Chronic obstructive airway disease		6 (0.9%)
Cerebrovascular disease		10 (1.5%)
Autoimmune disease		6 (0.9%)
Family history of diabetes mellitus		325 (49.2%)

**Table 4 medicina-60-00775-t004:** Differences in AUSDRISK by gender.

Items (N = 652)		Points	Overall	Female		Male		Test Statistics
				N	%	N	%	
Age	below 35 years	0	426 (65.3%)	223	74.1	203	57.8	χ^2^ = 21.963, *p* = 0.001
	35–44 years	2	129 (19.8%)	51	16.9	78	22.2	
	45–54 years	4	64 (9.8%)	18	6	46	13.1	
	55–64 years	6	29 (4.4%)	8	2.7	21	6	
	≥65 years	8	4 (0.6%)	1	0.3	3	0.9	
Have you ever been found to have high blood glucose (sugar)	No	0	579 (88.8%)	273	90.7	306	87.2	χ^2^ = 1.68, *p* = 0.195
	Yes	6	73 (11.2%)	28	9.3	45	12.8	
Have either of your parents or any of your brothers or sisters been diagnosed with diabetes	No	0	325 (49.8%)	157	52.2	157	44.7	χ^2^ = 7.527, *p* = 0.006
	Yes	3	327(50.2%)	194	64.5	194	55.3	
Are you currently taking medication for high blood pressure?	No	0	611 (93.7%)	295	98	316	90	χ^2^ = 16.18, *p* = 0.0001
	Yes	3	41 (6.3%)	6	2	35	10	
Do you currently smoke cigarettes or any other tobacco products daily?	No	0	485 (74.4%)	290	96.3	195	55.6	χ^2^ = 139.37, *p* = 0.0001
	Yes	2	167 (25.6%)	11	3.7	156	44.4	
How often do you eat vegetables or fruits?	No	0	138 (21.2%)	58	19.3	80	22.8	χ^2^ = 1.003, *p* = 0.316
	Yes	2	514 (78.8%)	243	80.7	271	77.2	
On average, would you say you do at least 2.5 h of physical activity per week?	No	0	311 (47.7%)	121	40.2	190	54.1	χ^2^ = 12.06, *p* = 0.001
	Yes	2	341 (52.3%)	180	59.8	161	45.9	
Waist circumference	Male < 102 cm/female < 88 cm	0	509 (78.1%)	260	86.4	249	70.9	χ^2^ = 27.00, *p* = 0.0001
	Male 102–110 cm/female 88–100 cm	4	83 (12.7%)	30	10	53	15.1	
	Male > 110 cm/female > 100	7	60 (9.2%)	11	3.7	49	14	
Total score			8.0 (5.0–31.0)	10.0 (7.0–16.0)		9	(7.0–14.0)	W = 16,294, *p* < 0.001

**Table 5 medicina-60-00775-t005:** Factors associated with higher AUSDRISK scores.

Dependent: Undiagnosed		AUSRISK ≥ 13	AUSDRISK < 13	*p*
Marital status	Divorced	8 (5.6)	16 (3.1)	<0.001
	Married	95 (66.4)	163 (32.0)	
	Single	37 (25.9)	327 (64.2)	
	Widow	3 (2.1)	3 (0.6)	
	Illiterate	0 (0.0)	5 (1.0)	
Education	Primary	1 (0.7)	0 (0.0)	<0.001
	Secondary	50 (35.0)	100 (19.6)	
	University	77 (53.8)	370 (72.7)	
	Postgraduate	15 (10.5)	34 (6.7)	
Income	Below 5000 SAR	36 (25.2)	231 (45.4)	<0.001
	5000–10,000 SAR	41 (28.7)	156 (30.6)	
	10,000–15,000 SAR	31 (21.7)	73 (14.3)	
	15,000–20,000 SAR	19 (13.3)	36 (7.1)	
	20,000 SAR	16 (11.2)	13 (2.6)	
Residence	Rural	26 (18.2)	42 (8.3)	0.001
	Urban	117 (81.8)	467 (91.7)	
Occupation	Do not work	26 (18.2)	226 (44.4)	<0.001
	Governmental sector	59 (41.3)	179 (35.2)	
	Private	32 (22.4)	95 (18.7)	
	Retired	26 (18.2)	9 (1.8)	
Nationality	Non-Saudi	11 (7.7)	35 (6.9)	0.879
	Saudi	132 (92.3)	474 (93.1)	
Ischemic heart disease	No	134 (93.7)	506 (99.4)	<0.001
	Yes	9 (6.3)	3 (0.6)	
Hyperlipidemia	Maybe	45 (31.5)	66 (13.0)	<0.001
	No	78 (54.5)	417 (81.9)	
	Yes	20 (14.0)	26 (5.1)	

## Data Availability

Data are available upon request upon emailing the corresponding author.

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
