# Peer review of "Prediction of Diabetes and Prediabetes among the Saudi Population Using a Non-Invasive Tool (AUSDRISK)"

_medicina, 2024, doi:10.3390/medicina60050775_

Round 1
Reviewer 1 Report
Comments and Suggestions for Authors Manuscript ID medicina-2964979
Dear Authors,
The introduction was well written. Indeed, non-invasive, cost effective tools would be more helpful in terms of mass screening of DM.
1. What was the reason for categorizing the income levels
2. There were more formatting errors, correct them.
3. In my opinion, AUSDRISK questionnaire seems inadequate, there should be some more questions to screen DM.
4. why "stress" was not included in the questionnaire,
asking this, since it plays a major role in DM.
5. Similarly, apart from vegetable intake what kind of diet was followed by the participants
6. How do you think this method will be favorable to screen DM
Comments on the Quality of English LanguageMinor changes in (in a few sentences) the writing style will be helpful.
Author Response
Reviewer #1
The introduction was well written. Indeed, non-invasive, cost-effective tools would be more helpful in the mass screening of DM.
Dear Reviewer # 1, We extend our heartfelt gratitude for your comprehensive review of our work. Your insights are invaluable and are certain to enhance both the content and scientific rigor of our study. We added a paragraph to show that AUSDRISK is a cost-effective tool.
[A team of Egyptian researchers translated the AUSDRISK into Arabic and assessed its reliability and validity among Egyptians. The re-calibrated Arabic AUSDRISK was shown to be a simple, reliable, and valid tool to predict diabetes risk, suggesting its use in mass public screening to reduce disease burden and health costs. [16] Therefore, its use in opportunistic mass public screening is highly recommended due to its cost-effectiveness. By employing this screening tool, it becomes possible to identify individuals at risk of developing DM2 early, which can significantly reduce the disease burden and associated health expenditure.]
Comment #1 What was the reason for categorizing the income levels
Thank you for your inquiry regarding the classification of income levels in our study. We primarily utilized data sourced from the Kingdom of Saudi Arabia | Monsha'at Business Atlas and other nationally published studies detailing income demographics. According to the data extracted from the atlas, the Eastern Province boasted the highest average household income at 14.9k SAR, whereas Najran reported the lowest at 8.7k SAR (reflecting a 58.4% difference). Based on these findings, we categorized income levels as follows: below 5000 SAR, 5000-10,000 SAR, 10,000-15,000 SAR, 15,000-20,000 SAR, and above 20,000 SAR per month. We hope this clarifies any questions you may have had regarding our methodology.
Reference [https://atlas.monshaat.gov.sa/en/profile/country/saudi-arabia]
Comment #2. There were more formatting errors, correct them.
Thank you for bringing the formatting errors to our attention. We are very sorry for any inconvenience caused. We have reviewed the document and made the necessary corrections to ensure clarity and consistency throughout. If you have any more concerns or suggestions, please don't hesitate to let us know.
Comment # 3. In my opinion, the AUSDRISK questionnaire seems inadequate, there should be some more questions to screen DM.
Dear Respected Reviewer, We sincerely appreciate your feedback and respect your opinion. It's important to note that the tool we utilized (AUSDRISK) has demonstrated significant sensitivity in detecting diabetes mellitus, as evidenced by previous studies where it has been extensively employed in its current form. Nevertheless, we acknowledge your suggestion that future studies could consider incorporating additional questions to further enhance the sensitivity and specificity of the tool. Your input is invaluable, and we will certainly take it into account for future research endeavors. Thank you once again for your insightful comments.
Here are some references that used AUSDRISK
- https://pubmed.ncbi.nlm.nih.gov/34762354/
- https://pubmed.ncbi.nlm.nih.gov/20854246/
- https://pubmed.ncbi.nlm.nih.gov/25707921/
- https://pubmed.ncbi.nlm.nih.gov/37231362/
- why "stress" was not included in the questionnaire, asking this, since it plays a major role in DM.
Thank you for your valuable observation. Indeed, the absence of stress assessment in our survey represents a notable limitation that warrants acknowledgment. Stress can undoubtedly influence various aspects of health, including those related to our study's focus. We appreciate your keen attention to detail and will ensure to address this limitation in our discussion of the study's findings. Your input is invaluable in refining the quality and thoroughness of our research.
[Finally, in this study, we did not focus on certain risk factors for the development of DM2, such as stress and dietary habits. Future studies may assess the value of incorporating these factors into the AUSDRISK tool to increase its diagnostic performance.]
- Similarly, apart from vegetable intake, what kind of diet was followed by the participants?
Dear Reviewer, Thank you for your attention to detail. We want to clarify that in our study, we solely relied on the questions from the AUSDRISK tool to predict the incidence of diabetes/prediabetes. Our analysis was based exclusively on the items included in the AUSDRISK questionnaire, without incorporating additional risk factors. It's important to note that the AUSDRISK tool includes only one question related to dietary habits, specifically asking about the frequency of vegetable and fruit consumption. While our study utilized this tool as it stands, we acknowledge the potential for future research to consider incorporating more questions into the AUSDRISK tool to enhance its predictive performance. We appreciate your feedback and will duly consider it for future investigations.
- How do you think this method will be favorable to screen DM
Dear Reviewer,
Thank you for your question regarding the potential effectiveness of our screening method for diabetes mellitus. Our approach utilizes the AUSDRISK tool, which has been validated and widely used for diabetes risk assessment. We believe that our method offers several advantages for screening DM. Firstly, the AUSDRISK tool is a validated invasive instrument specifically designed for diabetes risk prediction, ensuring reliability and accuracy in identifying individuals at risk. Additionally, our study demonstrates the practical application of this tool in a real-world setting, enhancing its clinical utility. Furthermore, by incorporating a simple and user-friendly questionnaire, our method enables easy implementation in various healthcare settings, including primary care facilities and community health screenings. This accessibility facilitates early detection and intervention, ultimately reducing the burden of undiagnosed diabetes and its associated complications. Overall, we are confident that our screening method holds promise in effectively identifying individuals at risk of DM, thereby enabling timely intervention and improving health outcomes.
Comments on the Quality of the English Language
Minor changes in (in a few sentences) the writing style will be helpful.
Dear Reviewer,
I appreciate your suggestion about minor adjustments to the writing style. We appreciate your feedback and have made the necessary changes to enhance the clarity and readability of this work.
Reviewer 2 Report
Comments and Suggestions for Authors
This is a well conducted, well presented and important study addressing prediction of diabetes and prediabetes in the Saudi population. There is some room for improvement.
1. It is unclear how many of the subjects had existing diabetes. How many were on treatment for diabetes or had self-reported diabetes? If so the results may require some rewriting.
2. In the introduction there should be a brief outline of Arabic version of AUSDRISK including the translation and validation so more than just quoting reference 16 - lines 78-84. You could include a simple table with the items and matching scores - maybe a simplified version of your table 3 (which you should retain).
3. Figures 1 and 2 and the text do not match. Please revise carefully.
4.The paragraph with lines 197-202 should be rewritten to make it clearer and easier to read. The aim is to convey the key findings and the reader can look at the table for the numerical details.
Suggest
More married than single participants had a high AUSDRISK score, p<.001.
Those with a University degree had lower scores, p< .001
Participants in the lowest income group had lower scores, p<.001.
5. There are some typographical errors
Line 105 AUSDRISK
Line 230 same issue
line 270 suggest "a non-invasive tool"
Comments on the Quality of English Language
Minor editing required
Author Response
Reviewer #2
Comments and Suggestions for Authors
This is a well-conducted, well-presented, and important study addressing the prediction of diabetes and prediabetes in the Saudi population. There is some room for improvement.
Thank you for your thoughtful feedback on our study. We're glad to hear that you found our research well-conducted, well-presented, and addressing an important topic regarding the prediction of diabetes and prediabetes in the Saudi population. We welcome any suggestions you may have for improvement and are committed to enhancing the quality and impact of our work.
Comment 1: It is unclear how many of the subjects had existing diabetes. How many were on treatment for diabetes or had self-reported diabetes? If so the results may require some rewriting.
Dear Reviewer, Thank you for your question. Indeed, as stated in the methodology section of our study, we specifically targeted apparently healthy Saudi adults aged 18 years or older residing in the Aseer region who did not have diabetes and were not using antidiabetic medications. We aimed to predict the incidence of diabetes within this population cohort. We appreciate your attention to detail and hope this clarifies any uncertainties you may have had regarding the study's scope and participant criteria.
Comment 2: In the introduction, there should be a brief outline of the Arabic version of AUSDRISK, including the translation and validation rather than just quoting reference 16 (lines 78–84). You could include a simple table with the items and matching scores - maybe a simplified version of your Table 3 (which you should retain).
Thank you for this comment. I have added more details about the Arabic version of the AUDSRISK in the introduction section. In the Methods, I added a simple table for each item and its score.
- Figures 1 and 2 and the text do not match. Please revise carefully.
Sorry for the inconvenience. The text was revised and corrected accordingly.
4.The paragraph with lines 197-202 should be rewritten to make it clearer and easier to read. The aim is to convey the key findings and the reader can look at the table for the numerical details.
We apologize once more for any inconvenience. I've rewritten this paragraph to better convey the main findings of the study.
Suggest
More married than single participants had a high AUSDRISK score, p<.001.
Rephrased
Those with a University degree had lower scores, (p<.001).
Rephrased
Participants in the lowest income group had lower scores, p<.001.
Rephrased
- There are some typographical errors
Line 105: AUSDRISK
Corrected
Line 230, same issue
Corrected
line 270 suggests "a non-invasive tool"
Corrected
Comments on the Quality of English Language
I appreciate your suggestion about minor adjustments to the writing style. We appreciate your feedback and have made the necessary changes to enhance the clarity and readability of this work.